# PROTEIN CAPTIONING: BRIDGING THE GAP BETWEEN PROTEIN SEQUENCES AND NATURAL LANGUAGES

## ABSTRACT

We introduce the task of **Protein Captioning**, which is an easy-to-understand and flexible way for protein analysis. Compared to specific protein recognition or classification tasks, such as enzyme reaction classification and gene ontology term prediction, protein captioning provides comprehensive textural descriptions for proteins, thus playing a key role in bridging the gap between protein sequences and natural languages. To address the problem, we propose a simple yet effective method, Protein-to-Text Generative Pre-trained Transformer (P2T-GPT), to translate the chain of amino acid residues in a protein to a sequence of natural language words, *i.e.*, text. For the evaluation of protein captioning, we collect a ProteinCap dataset that contains 94,454 protein-text pairs. Experiments on Protein-Cap demonstrate the effectiveness of the proposed P2T-GPT on protein captioning. As minor contributions, first, P2T-GPT provides a way to connect protein science and Large Language Models (LLMs). By appending ChatGPT, our method can interact in a conversational way to answer questions given a protein. Second, we show that protein captioning can be treated as a pre-trained task that can benefit a range of downstream tasks, to a certain extent. The code has been submitted in the supplementary material and will be publicly available.

## 1 INTRODUCTION

Proteins are large and complex biomolecules that play many critical roles in life. Understanding their function is essential for life-related sciences, including protein engineering, bioinformatics, drug design, medicinal chemistry, *etc*. Usually, it needs enormous biochemical experiments to find out proteins' function (Wüthrich, 2001; Jaskolski et al., 2014; Bai et al., 2015; Thompson et al., 2020). Recently, deep-learning-based approaches are developed for protein understanding, known as protein representation learning (Amidi et al., 2018; Kulmanov et al., 2018; Hou et al., 2018; Rao et al., 2019; Bepler & Berger, 2019; Alley et al., 2019; Strodthoff et al., 2020; Shanehsazzadeh et al., 2020; Kulmanov & Hoehndorf, 2021). However, existing protein representation learning methods usually focus on one or only a few specific and individual classification tasks, such as protein fold classification, enzyme reaction classification, gene ontology term prediction and enzyme commission number prediction. It is challenging for those methods to provide comprehensive descriptions of proteins. In contrast, natural language, as an effective medium for information expression, can provide more detailed descriptions and is easier to understand than task-specific predictions. In this paper, we introduce **Protein Captioning**, a new task that predicts the function, attribute, or other information of proteins via natural languages (shown in Figure 1).

Recently, Large Language Models (LLMs) (OpenAI, 2022; Brown et al., 2020a; Touvron et al., 2023; Zheng et al., 2023) have made remarkable advancements in natural language processing, demonstrating extraordinary reasoning ability and leading to an unprecedented language era. In this case, many research communities introduce LLMs into their own fields, which bring impressive improvements (Zhu et al., 2023; Li et al., 2023a; Dai et al., 2023; Wang et al., 2023b; Li et al., 2023b). However, in protein science, the integration of proteins with natural language is still in the early stages and has not been widely explored. Therefore, we propose a new problem that bridges the gap between protein sequences and natural languages, *i.e.*, protein captioning.

To address the protein captioning problem, based on Generative Pre-trained Transformer (GPT) (Brown et al., 2020b), we develop a simple yet effective model, named Protein-to-Text

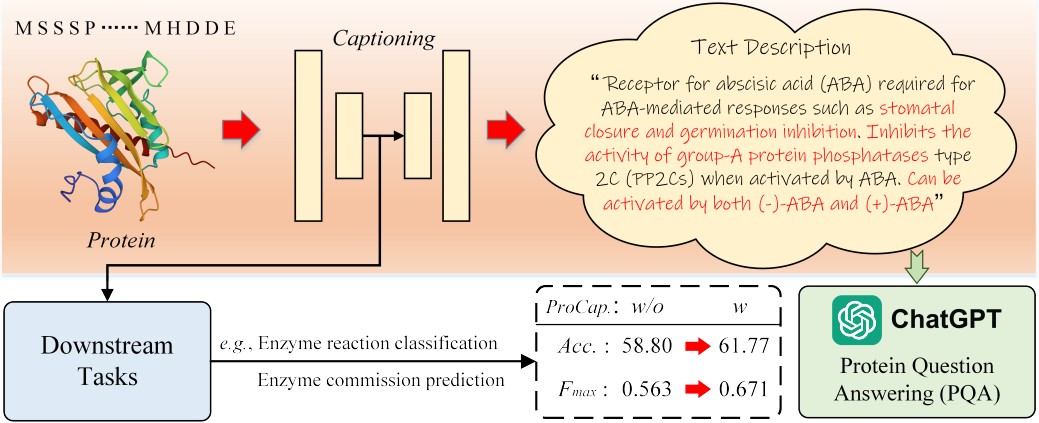

Figure 1: Illustration of protein captioning. Given a protein, the task generates a few natural language sentences to describe the type, function, source, or other information of the protein (highlight in red). By appending ChatGPT, our approach is able to facilitate protein question answering (a minor contribution). The protein captioning can also be treated as a pre-training strategy, aiding in a range of downstream tasks (a minor contribution).

GPT (P2T-GPT). Specifically, P2T-GPT consists of a protein encoder and a causal decoder. To effectively and efficiently model protein sequences, our encoder first employs a Convolutional Neural Network (CNN) to encode the short-range dependency of amino acids and then stacks multiple Transformer blocks (Vaswani et al., 2017b) to capture the long-range relationship among protein regions or segments. For the causal decoder, a GPT-like architecture is used to generate words based on the encoded protein representation. Moreover, we integrate a residue-word cross-attention mechanism into the causal decoder Transformer to align the protein-specific information and the corresponding descriptions.

To train and evaluate protein captioning models, we collect a large-scale dataset, named ProteinCap. The dataset contains about 94k protein-text pairs and involves a range of various proteins, from different species and with different functions. On the ProteinCap dataset, we demonstrate that our approach can generate reasonable descriptions for proteins. Furthermore, as a minor contribution, by appending ChatGPT, our method can interact in a conversational way to answer questions given a protein. We also show that, by pre-training protein encoders with captioning, downstream tasks (*e.g.*, enzyme reaction classification) are improved, indicating that protein captioning can be used as a pre-training task. The contributions of this paper are fivefold:

- We introduce the protein captioning task. Compared to classification-based protein representation learning tasks, protein captioning provides an easy-to-understand and flexible way for protein analysis.
- We propose a P2T-GPT framework for protein captioning. P2T-GPT can effectively model the short-range and long-range dependencies of amino acids and translate protein sequences into comprehensive textual descriptions.
- We collect a large-scale ProteinCap dataset, which contains more than 94k protein-text pairs, for training and evaluating protein captioning.
- By appending ChatGPT, our method can be used for protein question answering (a minor contribution).
- Protein captioning can be used as a pre-training strategy that is able to improve downstream tasks (a minor contribution).

## 2 RELATED WORK

**Protein Representation Learning.** Research on protein representation learning has a long history (Murzin et al., 1995). Recently, deep-learning-based artificial intelligence becomes a widespread solution for protein modeling (*e.g.*, protein structure classification and function classification), leading to a better understanding of structural bioinformatics. One intuitive method is to model 1D amino acid sequences via CNN, LSTM, and Transformer (Shanehsazzadeh et al., 2020; Rao et al., 2019) in a fully supervised manner. Inspired by natural language models, many works explore self-supervised

protein representation pre-training via Masked Language Modeling (MLM). These works focus on enlarging datasets (Elnaggar et al., 2021; Rives et al., 2021), investigating different architectures (Rao et al., 2021; Vig et al., 2021; Yang et al., 2022; Chen et al., 2023a), prompt learning (Wang et al., 2023c), and introducing extra knowledge base (Zhang et al., 2022a; Zhou et al., 2023). Besides, some approaches aim to obtain higher-quality protein representations by using 3D geometry information (Kipf & Welling, 2017; Derevyanko et al., 2018; Wang et al., 2023a; Fan et al., 2023a;b; Chen et al., 2023b) or introducing both amino acid 1D sequences and 3D coordinates (Baldassarre et al., 2021; Hermosilla & Ropinski, 2022; Zhang et al., 2022b; Fan et al., 2023a). In this paper, because most protein databases only provide the primary structure, we focus on generating textual descriptions based on protein sequences.

**Visual Captioning.** Connecting vision and language plays an essential role in artificial intelligence. Visual captioning, which aims at describing the content of an image or a video in words, lies at the intersection of computer vision and natural language processing. Usually, visual captioning consists of a visual encoder to extract vision representations and a language decoder to generate textual descriptions. For visual encoding, early-proposed approaches are based on global CNN features (Vinyals et al., 2015; Mao et al., 2015; Donahue et al., 2015; Chen & Zitnick, 2015; Fang et al., 2015; Jia et al., 2015). This paradigm leads to excessive compression of information and lacks granularity, making it hard for a captioning model to produce specific and fine-grained descriptions. To address this problem, attention-based methods are proposed to increase the granularity level of visual encoding (Lu et al., 2017; Dai et al., 2018; Yang et al., 2016). In particular, self-attention or Transformer (Vaswani et al., 2017a) recently are widely used as visual encoder to compute a refined visual representation (Yang et al., 2019; Guo et al., 2020; Huang et al., 2019). For language decoding, RNN variants, such as Long short-term memory (LSTM) (Hochreiter & Schmidhuber, 1997) and Gated Recurrent Unit (GRU) (Chung et al., 2014), have been the predominant option for language modeling. Then, Transformer-based architectures become the fundamental components of natural language processing, especially Large Language Models (LLMs) (OpenAI, 2022; Touvron et al., 2023; Zheng et al., 2023; Driess et al., 2023). Various methods (Liu et al., 2023; Zhu et al., 2023; Li et al., 2023a; Dai et al., 2023; Wang et al., 2023b; Li et al., 2023b; Zhang et al., 2023a) generate comprehensive responses by incorporating visual encoders and LLMs, which completely change the perspective of language generation. Inspired by those Transformer-based methods, we propose a P2T-GPT model for protein captioning.

## 3 PROTEIN CAPTIONING TASK AND DATASET

### 3.1 TASK STATEMENT

The task of protein captioning aims to generate textual descriptions of protein functions. Specifically, given an amino acid sequence $[a_1, a_2, \cdots, a_N]$, where $a_i \in \{1, \cdots, 21\}$ is the type of the $i$-th amino acid and $N$ is the number of amino acids in the protein, protein captioning produces a human-like description $[w_1, w_2, \cdots, w_T]$, where $w_i \in \{1, \cdots, M\}$ is the ID of the $i$-th word token, $M$ is the size of the vocabulary and $T$ is the number of tokens in the description.

The model is expected to encode the amino acid sequence and describe its function, attribute, or other information. The quality of the generated function can be improved by encouraging consistency between the generated text and the ground truth.

Table 1: Due to the small size of PubMedBERT's vocabulary (Gu et al., 2020), many keywords of proteins' functional descriptions are excluded. Those missing keywords have to be replaced with the [UNK] token. We rebuild a new vocabulary that includes all keywords.

**Tokenized description based on PubMedBERT's vocabulary:**
[UNK] [UNK] potassium channels ([UNK]), [UNK] potassium channel ([UNK]), and the calcium release [UNK] receptor ([UNK]).

**Ours:**
Inhibits calcium-activated potassium channels (KCa), voltage-gated potassium channel (Kv), and the calcium release channel/ryanodine receptor (RyR).

### 3.2 PROTEINCAP DATASET

**Dataset collection.** ProteinCap dataset is collected from 569,213 proteins in the Swiss-Prot dataset, which can be found in the publicly available database, UniProt[1]. It contains proteins from a wide range of organisms, such as the Human, Mouse, A.thaliana, *etc*, with comprehensive properties.

---

[1] https://www.uniprot.org/

Table 2: Specification of different splits of the ProteinCap dataset.

| Name | # Residues ($N$) | # Words ($T$) | Train | Val | Test | Total | Vocab. Size |
|---|---|---|---|---|---|---|---|
| ProteinCap-$\alpha$ | $\{20 \le N \le 200\}$ | $\{T \le 100\}$ | 75,563 | 9,445 | 9,446 | 94,454 | 28,860 |
| ProteinCap-$\beta$ | $\{20 \le N \le 200\}$ | $\{T \le 50\}$ | 59,982 | 7,497 | 7,499 | 74,978 | 19,320 |
| ProteinCap-$\gamma$ | $\{20 \le N \le 100\}$ | $\{T \le 50\}$ | 15,208 | 1,901 | 1,901 | 19,010 | 10,022 |

In this work, we choose "primaryAccession" (protein ID), "length" (protein length), "sequence" (amino acid sequence), and "function" (text descriptions of protein's function, attribute, or other information) to build our dataset for protein captioning. Previous protein representation learning works (Wang et al., 2023c; Zhou et al., 2023) use text vocabulary provided in PubMedBERT (Gu et al., 2020) for tokenizing. However, PubMedBERT vocabulary does not contain all words in the function description, leading to incorrect or unreasonable results. For example in Table 1, the tokenized sentence cannot describe the protein due to core words (*i.e.*, `channel/ryanodine` and `calcium-activated`) are replaced with `[UNK]`. Thus, we build a new vocabulary collected from all functional descriptions to describe proteins precisely.

**Dataset filtering.** Because many samples have exactly the same protein sequences and text descriptions in the originally collected dataset, we remove the repeated samples and keep only one of them. Moreover, we filter the dataset with both textual length and amino acid sequence length as thresholds. We select the protein-text pairs that satisfy the requirements of both protein sequence length $N \in [20, 200]$, and text length $T \in [0, 100]$.

**Dataset split.** The ProteinCap dataset contains 94,454 filtered protein-text pairs in total. We constructed three subsets based on the length of protein sequence and text, and then split them into the training set, validation set, and testing set under the 8:1:1 partition protocol, respectively. The detailed information of the three subsets is shown in Table 2.

## 4 METHOD

In this paper, we propose a P2T-GPT that is able to generate reasonably functional descriptions for protein sequences. As shown in Figure 2, P2T-GPT consists of a protein encoder and a causal captioning decoder. In Section 4.1, we present the protein encoder that captures the amino acid sequence structure in a local-to-global manner. In Section 4.2, we introduce the causal captioning decoder that is responsible for generating the textual description based on the protein feature. In Section 4.3, we provide implementation details.

### 4.1 PROTEIN ENCODER

The goal of the protein encoder is to learn the effective representations for proteins. Specifically, our protein encoder consists of an amino acid embedding layer, a convolutional neural network, and a Transformer. First, the embedding layer converts an amino acid type into a vector $\boldsymbol{a} \in \mathbb{R}^{1 \times C_a}$. Then, the convolutional neural network is used to capture the local structure of amino acid sequences. Given the embedded amino acid vectors $\boldsymbol{A} = [\boldsymbol{a}_1; \boldsymbol{a}_2; \cdots; \boldsymbol{a}_N] \in \mathbb{R}^{N \times C_a}$, convolution captures the local structure as follows,

$$\boldsymbol{a}_i' = \sum_{\delta=-\lfloor K/2 \rfloor}^{\lfloor K/2 \rfloor} \boldsymbol{W}_\delta \cdot \boldsymbol{a}_{i+\delta}^\top,$$ (1)

where $K$ is the size of kernel or receptive field, $\boldsymbol{W}_\delta \in \mathbb{R}^{C_a \times C_a'}$ is the learnable parameters and "·" is the matrix multiplication. During performing convolution, we downsample residues with a rate $r \in (0, 1)$. Suppose the network contains $m$ convolutional layers, there are $N' = r^m N$ residues after the downsampling, leading to the output $\boldsymbol{A}' \in \mathbb{R}^{N' \times C_a'}$. Third, to model the long-range dependency in proteins, we employ Transformer with the vanilla self-attention as follows,

$$\boldsymbol{P} = \texttt{Softmax}\left(\frac{\boldsymbol{Q}_a \cdot \boldsymbol{K}_a^\top}{\sqrt{C_a''}}\right) \cdot \boldsymbol{V}_a,$$ (2)

where $\boldsymbol{Q}_a = \boldsymbol{A}' \cdot \boldsymbol{W}_q^a$, $\boldsymbol{K}_a = \boldsymbol{A}' \cdot \boldsymbol{W}_k^a$, $\boldsymbol{V}_a = \boldsymbol{A}' \cdot \boldsymbol{W}_v^a$, and $\boldsymbol{W}_q^a \in \mathbb{R}^{C_a' \times C_a''}$, $\boldsymbol{W}_k^a \in \mathbb{R}^{C_a' \times C_a''}$, $\boldsymbol{W}_v^a \in \mathbb{R}^{C_a' \times C_a''}$. In this way, the network encodes a protein to $\boldsymbol{P} \in \mathbb{R}^{N' \times C_a''}$, which are then used for text generation.

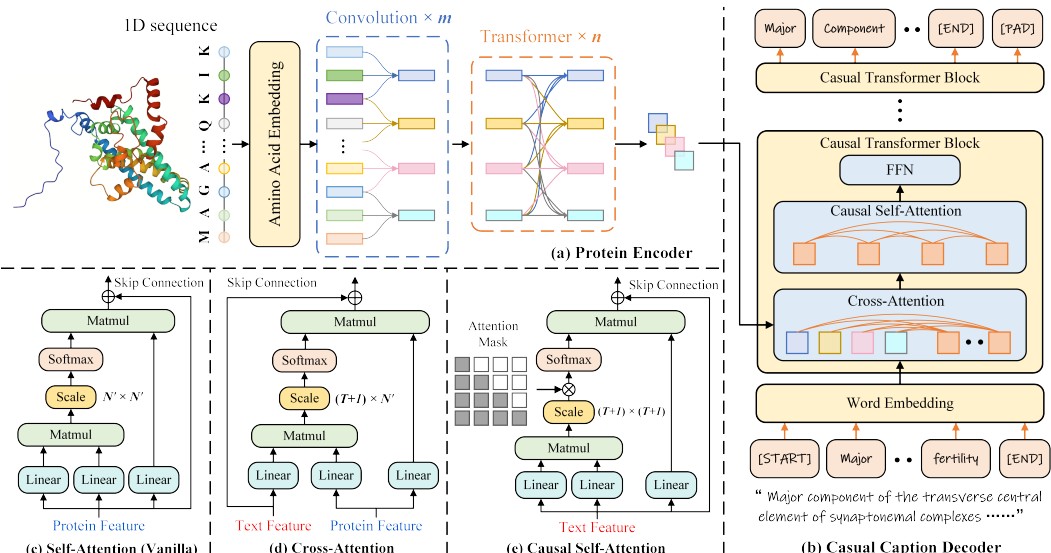

Figure 2: Illustration of the P2T-GPT architecture. The protein encoder first employs a CNN to encode the short-range dependency of amino acids and then stacks Transformers to capture the long-range relationship among protein regions. The causal caption decoder employs a GPT-like architecture to generate words based on the encoded protein representation.

## 4.2 CAUSAL CAPTIONING DECODER

Based on the extracted protein feature by the protein encoder, the causal captioning decoder generates the functional text description in an autoregressive fashion. First, we add an extra [START] token and a [END] token as a signal to start and stop the generation, respectively. Specifically, the textual description $[w_1, w_2, \cdots, w_T]$ is extended as $[\text{START}, w_1, w_2, \cdots, w_T]$ for input and $[w_1, w_2, \cdots, w_T, \text{END}]$ for output. Then, a word embedding layer converts each token into a word representation $\boldsymbol{w} \in \mathbb{R}^{1 \times C_t}$. Given the embedded word representation $\boldsymbol{T} = [\boldsymbol{w}_1; \boldsymbol{w}_2; \cdots; \boldsymbol{w}_T] \in \mathbb{R}^{(T+1) \times C_t}$, we aim to project $\boldsymbol{T}$ back to $[w_1, w_2, \cdots, w_T, \text{END}]$ through the next-token prediction based on protein representation. This process can be formulated as $p(w_i | \boldsymbol{P}, \boldsymbol{T}_{<i})$.

The architecture of the causal captioning decoder is shown in Figure 2(b). It consists of a cross-attention module and a causal attention module. The cross-attention component aims to attend the correct protein segment when generating the corresponding function. Specifically, cross-attention is formulated as follows,

$$\boldsymbol{T}' = \texttt{Softmax}\left(\frac{\boldsymbol{Q}_t \cdot \boldsymbol{K}_a'^\top}{\sqrt{C_t'}}\right) \cdot \boldsymbol{V}_a', \tag{3}$$

where $\boldsymbol{Q}_t = \boldsymbol{T} \cdot \boldsymbol{W}_q^t$, $\boldsymbol{K}_a' = \boldsymbol{P} \cdot \boldsymbol{W}_k^{a'}$, $\boldsymbol{V}_a' = \boldsymbol{P} \cdot \boldsymbol{W}_v^{a'}$, $\boldsymbol{W}_q^t \in \mathbb{R}^{C_t \times C_t'}$, $\boldsymbol{W}_k^{a'} \in \mathbb{R}^{C_a'' \times C_t'}$ and $\boldsymbol{W}_v^{a'} \in \mathbb{R}^{C_a'' \times C_t'}$. In this way, the network produces a cross-attended representation $\boldsymbol{T}' \in R^{(T+1) \times C_t'}$, which is then used for the next-word prediction.

Then, causal self-attention is employed for textural description generation. Specifically, when predicting the $i$-th token, only the previous $i-1$ words can be seen. This process can be as a masked self-attention mechanism and formulated as follows,

$$\boldsymbol{T}'' = \texttt{Softmax}\left[\frac{\mathcal{D}(\boldsymbol{Q}_t' \cdot \boldsymbol{K}_t^\top, \mathcal{M})}{\sqrt{C_t''}}\right] \cdot \boldsymbol{V}_t, \tag{4}$$

where $\boldsymbol{Q}_t' = \boldsymbol{T}' \cdot \boldsymbol{W}_q^{t'}$, $\boldsymbol{K}_t = \boldsymbol{T}' \cdot \boldsymbol{W}_k^t$, $\boldsymbol{V}_t = \boldsymbol{T}' \cdot \boldsymbol{W}_v^t$, $\boldsymbol{W}_q^{t'} \in \mathbb{R}^{C_t' \times C_t''}$, $\boldsymbol{W}_k^t \in \mathbb{R}^{C_t' \times C_t''}$ and $\boldsymbol{W}_v^t \in \mathbb{R}^{C_t' \times C_t''}$. In this way, the network generates the predicted token representation $\boldsymbol{T}'' \in \mathbb{R}^{(T+1) \times C_t''}$. The function $\mathcal{D}$ masks the attention map through the indicator $\mathcal{M}$ as follows,

$$\mathcal{D}(x) = \begin{cases} x, & \mathcal{M}_{i,j} = 1 \\ -\infty, & \mathcal{M}_{i,j} = 0 \end{cases} \quad i, j \in [1, 2, ..., T, T+1]. \tag{5}$$

Table 3: Quantitative results on the ProteinCap dataset under three partition protocol. 'CA' and 'SA' denote the cross-attention-based and self-attention-based P2T-GPT, respectively.

| Data | | BLEU (%) ↑ | | | | BERTScore ↑ (%) | ROUGE-L ↑ (%) | METEOR ↑ (%) | CIDEr ↑ |
| | | BLEU@1 | BLEU@2 | BLEU@3 | BLEU@4 | | | | |
|---|---|---|---|---|---|---|---|---|---|
| ProteinCap-$\alpha$ | SA | 30.95 | 27.56 | 25.78 | 24.77 | 22.00 | 34.19 | 22.36 | 1.82 |
| | CA | 82.87 | 81.43 | 80.70 | 80.24 | 77.73 | 81.72 | 76.36 | 7.55 |
| ProteinCap-$\beta$ | SA | 48.26 | 43.40 | 40.70 | 39.04 | 36.04 | 47.61 | 37.73 | 3.49 |
| | CA | 83.41 | 82.22 | 81.65 | 81.29 | 80.15 | 83.13 | 79.17 | 7.84 |
| ProteinCap-$\gamma$ | SA | 68.25 | 66.10 | 64.77 | 63.77 | 48.95 | 62.59 | 50.66 | 4.75 |
| | CA | 76.70 | 75.39 | 74.80 | 74.46 | 72.63 | 75.95 | 72.09 | 6.99 |

where $\mathcal{M}_{i,j} = 1$ if $i \geq j$ and $\mathcal{M}_{i,j} = 0$ if $i < j$.

Last, a Multi-Layer Perceptron (MLP) is used to project $\boldsymbol{T}''$ to the final prediction $\boldsymbol{O} = \texttt{MLP}(\boldsymbol{T}'') \in \mathbb{R}^{(T+1)\times M}$, where $\boldsymbol{O}_i \in \mathbb{R}^{1\times M}$ is the prediction probability over the vocabulary of the $i$-th token. The index of the maximum value $\boldsymbol{O}_i$ will be treated as the prediction result for the current word token, *i.e.*, $\hat{w}_i = \texttt{argmax}(\boldsymbol{O}_i)$.

During inference, the captioning starts from the [START] token and iteratively generates descriptions word by word. The generation process will stop when the [END] token is generated in the prediction.

### 4.3 IMPLEMENTATION DETAILS

**Optimization.** During training, our goal is to conduct consistent regularization between $[\texttt{START}, w_1, w_2, \cdots, w_T]$ and $[w_1, w_2, \cdots, w_T, \texttt{END}]$. To optimize the model, we maximize the negative log-likelihood distribution with cross-entropy,

$$\mathcal{L} = \mathbb{E}[-\log \prod_{i=1}^{T+1} p(w_i|\boldsymbol{P}, \boldsymbol{T}_{<i})] \tag{6}$$

To train in a mini-batch manner, we pad protein sequences and text sequences to the same length with the [PAD] token.

**Network architecture.** For all experiments, we set the dimension of the protein embedding $C_a$, $C_a'$, and $C_a''$ to 256, 512, and 512, and text embedding dimension $C_t = C_t' = C_t''$ to 512, respectively. In the protein encoder, we set the downsampling rate $r = 0.5$, $m = 4$, and the convolutional kernel size $K$ to 15. We stack $n = 4$ transformers with dimensions of $C_a''$ and 8 heads. As to the causal caption decoder Transformer, the number of layers is set to 8 and the dropout rate to 0.1. The embedding dimensionality is set to 512 and the number of heads is set to 8.

## 5 EXPERIMENTS

### 5.1 PROTEIN CAPTIONING

**Dataset.** The evaluation for protein captioning is carried out on ProteinCap-$\alpha$, ProteinCap-$\beta$, and ProteinCap-$\gamma$ (for more details, see Section 3.2). Note that we provide a diagnostic study on dataset size in Appendix C.

**Evaluation metric.** Our method is measured with the following five evaluation metrics: BLEU@1-4 (Papineni et al., 2002), BERTScore (Zhang et al., 2019), ROUGE-L (Lin & Och, 2004), METEOR (Banerjee & Lavie, 2005), and CIDEr (Vedantam et al., 2015). We select the model which achieves the best BLEU-1 on the validation set, and then evaluate it on the test set. More details are provided in Appendix A.

**Training.** All models are trained on $4 \times$ NVIDIA RTX A4000 GPUs by AdamW (Loshchilov & Hutter, 2019) optimizer with batch size 64. For ProteinCap-$\alpha$ and ProteinCap-$\beta$, we initialize the learning rate as $1 \times 10^{-4}$ for 100K iterations and decayed to $1 \times 10^{-5}$ for another 50K iterations. As to the ProteinCap-$\gamma$, we train 100K iterations in total, and the learning rate change from $1 \times 10^{-4}$ to $1 \times 10^{-5}$ after 60K iterations.

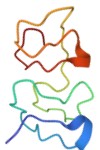

*Protein ID: P80258*

**Ground Truth：** The metallothioneins are involved in the cellular sequestration of toxic metal ions.

**P2T-GPT：** The metallothioneins are involved in the cellular sequestration of toxic metal ions.

**Baseline:** Metallothioneins have a high content of cysteine residues that bind various heavy metals.

---

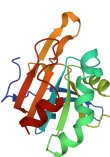

*Protein ID: O22711*

**Ground Truth：** Thiol-specific peroxidase that catalyzes the reduction of hydrogen peroxide and organic hydroperoxides to water and alcohols , respectively . Plays a role in cell protection against oxidative stress by detoxifying peroxides ( By similarity ) . May be involved in intracellular redox signaling ( Probable ).

**P2T-GPT :** Thiol-specific peroxidase that catalyzes the reduction of hydrogen peroxide and organic hydroperoxides to water and alcohols , respectively . Plays a role in cell protection against oxidative stress by detoxifying peroxides and as sensor of hydrogen peroxide-mediated signaling events.

**Baseline:** Catalyzes the formation of the biosynthesis of a beta-alanine.

---

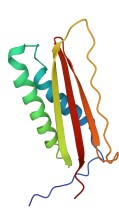

*Protein ID: Q8SPS9*

**Ground Truth：** Acts as one of several non-catalytic accessory components of the cytoplasmic dynein 1 complex that are thought to be involved in linking dynein to cargos and to adapter proteins that regulate dynein function . Cytoplasmic dynein 1 acts as a motor for the intracellular retrograde motility of vesicles and organelles along microtubules . Probably binds BUB3 as part of transport cargo . Required for the efficient progression through mitosis ( By similarity )

**P2T-GPT :** Acts as one of several non-catalytic accessory components of the cytoplasmic dynein 1 complex that are thought to be involved in linking dynein to cargos and to adapter proteins that regulate dynein function . Cytoplasmic dynein 1 acts as a motor for the intracellular retrograde motility of vesicles and organelles along microtubules . Probably binds lymphoma transport . Probably binds to transport in apical adapter proteins , such as other reductase , synthesis , form factors.

**Baseline:** Involved in the hydrolysis of the presence of the cell.

Figure 3: Qualitative comparison. We compare our method with the baseline that does not employ cross-attention. The blue and red colors indicate correct and incorrect descriptions, respectively. Our P2T-GPT generates reasonable results, especially for simple proteins with short textual descriptions. More qualitative results are provided in Appendix F.

**Quantitative results.** We show quantitative results in Table 3 on ProteinCap-$\alpha, \beta, \gamma$ test sets. 'SA' is the baseline model, which stacks the protein feature and embedded text feature together for training (The architecture of the baseline is provided in Appendix D). On all datasets, our approach outperforms the self-attention baseline for all evaluation metrics. We find that the performance degrades significantly as the length of amino acid and functional text description become longer for the baseline model, while our cross-attention-based approach solves this problem. For example, our method obtains an improvement of 55.47% (BLEU-4), 47.53% (ROUGE-L), 54.00% (METEOR), and 5.73 (CIDEr) on ProteinCap-$\alpha$ compared with the self-attention baseline, respectively. Based on the pre-trained language model, the BERTScore provides a better understanding of the semantics of textual descriptions than n-gram-based methods, *e.g.*, BLEU. For BERTScore, our method outperforms the baseline method by 55.73% on the ProteinCap-$\alpha$ dataset.

**Qualitative results.** Figure 3 shows the qualitative comparison between P2T-GPT and the baseline on ProteinCap-$\alpha, \beta, \gamma$ test sets. The blue and red colors indicate correct and incorrect descriptions, respectively. P2T-GPT generates similar functional descriptions as ground truth, but the baseline generates incorrect descriptions. Besides, Figure 4 shows the cross-attention map of P2T-GPT on the ProteinCap-$\alpha$ test set. We select two different amino acid sequences with the same functional descriptions from ProteinCap-$\alpha$. It can be seen that amino acid sequences corresponding to the highly activated text region are similar, which proves the effectiveness of our approach. Note that 3D protein structures are visualized by Protein Viewer (Sehnal et al., 2021), and more qualitative results and failure cases are provided in Appendix F.

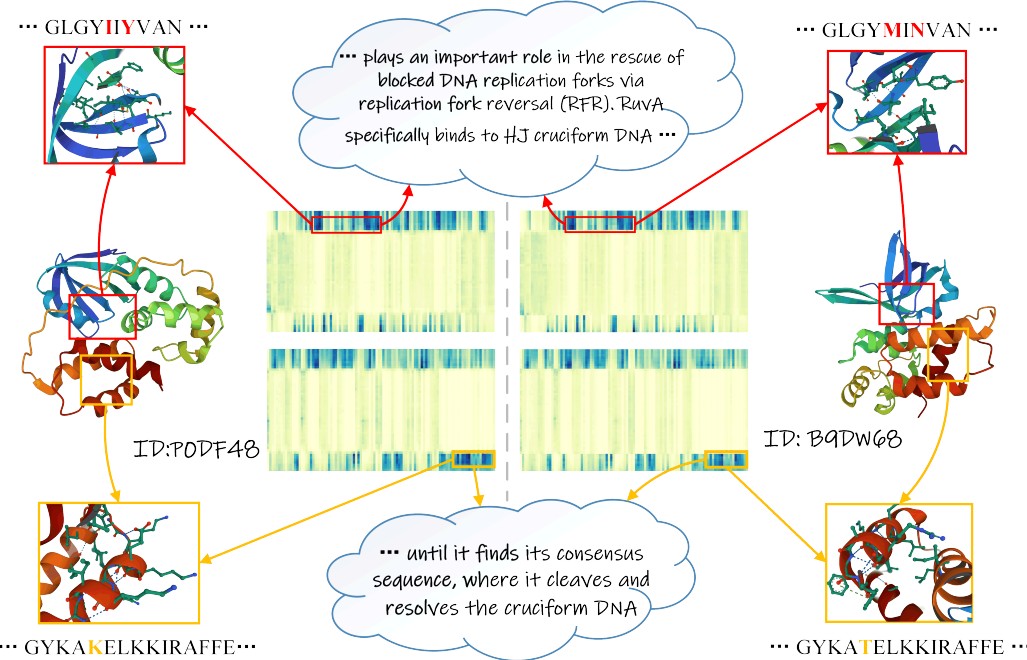

Figure 4: Visualization of the cross-attention map. Results are from the ProteinCap-$\alpha$ test dataset. We compare two different protein sequences (P0DF48 and B9DW68) with the same functional descriptions. As expected, the attended regions of the two proteins are similar for the same function. This demonstrates our cross-attention is able to attend to the correct protein segment when generating the corresponding functional description. More results are shown in Appendix F.

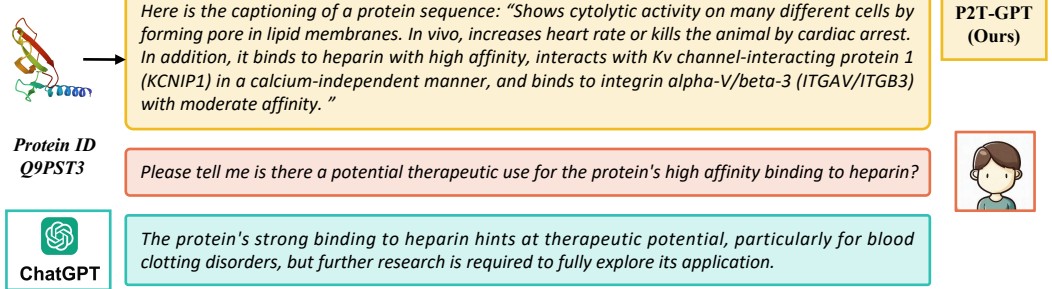

Figure 5: Illustration of combining P2T-GPT and ChatGPT for Protein Question Answering (PQA). Given an uploaded protein and a user question, P2T-GPT converts the protein to a textural caption and then ChatGPT answers the question based on the generated caption (a minor contribution).

## 5.2 PROTEIN QUESTION ANSWERING

Protein Question Answering (PQA) is an extension of the captioning task. It provides a more flexible way than captioning to obtain the most informative knowledge of interest. In this section, we demonstrate how to combine our P2T-GPT with ChatGPT for PQA. We show the conversation of ChatGPT (OpenAI, 2022) answering the question in Figure 5. Given the user question, ChatGPT is able to generate reasonable answers based on the protein captioning generated by P2T-GPT. We provide multiple-round question-answer processes in Appendix F.

## 5.3 PRE-TRAINING FOR DOWNSTREAM TASKS

In this section, we show that treating protein captioning as a pre-training strategy improves the performance of P2T-GPT's protein encoder across various downstream tasks.

**Datasets.** We conduct experiments on four widely used protein recognition tasks: Protein Fold Classification (Hou et al., 2018), Enzyme Reaction Classification (Hermosilla et al., 2021), Gene Ontology

Table 4: Comparison with existing sequence-based methods on four downstream tasks (a minor contribution). Mean accuracy (%) is used for evaluating protein fold classification and enzyme reaction classification. We compute $F_{max}$ for gene ontology term prediction and enzyme commission number prediction. "w/o ProteinCap" denotes the experiments are training from scratch without pre-training on ProteinCap. "ESM-2" indicates that we finetune the pre-trained ESM-2 (Lin et al., 2023) on four downstream tasks, "ESM-2 + Ours (w/o ProteinCap)" and "ESM-2 + Ours (w/ ProteinCap)" denote that we combine pre-trained ESM-2 following ESM-GearNet (Zhang et al., 2023b) with our plain and ProteinCap pre-trained protein encoder, respectively. The [§] indicates results are from (Fan et al., 2023a).

| Method | Fold Classification | | | Enzyme | Gene Ontology | | | Enzyme |
| | Fold | Superfamily | Family | Reaction | BP | MF | CC | Commission |
|---|---|---|---|---|---|---|---|---|
| CNN (Shanehsazzadeh et al., 2020)[§] | 11.3 | 13.4 | 53.4 | 51.7 | 0.244 | 0.354 | 0.287 | 0.545 |
| ResNet (Rao et al., 2019)[§] | 10.1 | 7.21 | 23.5 | 24.1 | 0.280 | 0.405 | 0.304 | 0.605 |
| LSTM (Rao et al., 2019)[§] | 6.41 | 4.33 | 18.1 | 11.0 | 0.225 | 0.321 | 0.283 | 0.425 |
| Transformer (Rao et al., 2019)[§] | 9.22 | 8.81 | 40.4 | 26.6 | 0.264 | 0.211 | 0.405 | 0.238 |
| Ours (w/o ProteinCap) | 12.12 | 13.00 | 67.53 | 58.80 | 0.292 | 0.375 | 0.387 | 0.563 |
| Ours (w/ ProteinCap) | **14.07** | **18.98** | **78.77** | **61.77** | **0.313** | **0.416** | **0.419** | **0.671** |
| ESM-2 (Lin et al., 2023) | 24.51 | **49.92** | 93.95 | 79.84 | 0.368 | 0.544 | 0.409 | 0.781 |
| ESM-2 + Ours (w/o ProteinCap) | 26.18 | 47.13 | 93.24 | 79.01 | 0.377 | 0.543 | 0.409 | 0.780 |
| ESM-2 + Ours (w/ ProteinCap) | **28.83** | 49.28 | **94.50** | **81.26** | **0.382** | **0.548** | **0.436** | **0.786** |

Term Prediction (Gligorijević et al., 2021) and Enzyme Commission Number Prediction (Gligorijević et al., 2021). More details are provided in Appendix B.

**Evaluation metric.** Following (Rao et al., 2019; Shanehsazzadeh et al., 2020; Gligorijević et al., 2021), protein fold classification and enzyme reaction classification are measured by mean accuracy, and $F_{max}$ is used for gene ontology term prediction and enzyme commission number prediction evaluation. See Appendix A for more details.

**Training.** We introduce a global token and attach a linear project head on top of the global token, and train the project head and the protein encoder simultaneously. All tasks are trained on a single NVIDIA RTX A4000 GPU with AdamW (Loshchilov & Hutter, 2019) optimizer and a batch size of 128. More details of implementation and training setup are provided in Appendix B.

**Comparison to state-of-the-arts.** We compare our approach with existing 1D-only models, *i.e.* CNN (Shanehsazzadeh et al., 2020), ResNet (Rao et al., 2019), LSTM (Rao et al., 2019) and Transformer (Rao et al., 2019). We use the model that is pre-trained on ProteinCap-$\alpha$. Experimental results in Table 4 demonstrate that, with protein captioning, our method achieves a certain degree of improvement.

Our approach can further enhance the state-of-the-art pre-trained model, *i.e.*, ESM-2 (Lin et al., 2023). Following ESM-GearNet (Zhang et al., 2023b), we fuse the predictions of ESM-2 and our method. As shown in Table 4, our method improves the accuracy.

Besides, it can be seen that the performance of the previous method fluctuates on different tasks. For example, CNN (Shanehsazzadeh et al., 2020) obtains the highest accuracy on enzyme reaction classification, while ResNet (Rao et al., 2019) performs best on enzyme commission number prediction. On the other hand, our method achieves consistent improvements on all tasks.

## 6 CONCLUSION

In this work, we first introduce a novel task of protein captioning, which provides an easy-to-understand and flexible way for protein analysis. For this task, we build a dataset that comprises 94,454 protein-text pairs, named ProteinCap. Then, we propose a P2T-GPT framework for protein captioning. P2T-GPT consists of a protein encoder to capture both local structure and long-range dependency of amino acid sequences, and a causal captioning decoder to synthesize text descriptions from protein features in an autoregressive fashion. Furthermore, we demonstrate that protein captioning is able to facilitate protein question answering by appending ChatGPT. Finally, protein captioning can also be treated as a pre-training strategy to benefit downstream tasks.

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

## APPENDIX OVERVIEW

This appendix includes:

We also provide the code in the supplementary material.

## A  DETAILS OF EVALUATION METRICS

For protein captioning:

- *BERTScore* (Papineni et al., 2002) measures semantic-level similarity which leverages the pre-trained contextual embeddings from BERT. It evaluates the performance via cosine similarity between the generation and ground truth, which is more highly correlated with human judgment than existing metrics, and provides enhanced model selection performance.
- *Bilingual Evaluation Understudy (BLEU)* (Papineni et al., 2002) measures the caption precision between the generated sentence and ground truth. We use BLEU-1, BLEU-2, BLEU-3, and BLEU-4 under $N$-gram ($N \in [1, 2, 3, 4]$) evaluation.
- *Recall-Oriented Understudy for Gisting Evaluation (ROUGE)* (Lin & Och, 2004) is similar to BLEU which compares the generated sentence against ground truth with recall. We take ROUGE-L for evaluation based on the Longest Common Sub-sequence.
- *Metric for Evaluation of Translation with Explicit Ordering (METEOR)* (Banerjee & Lavie, 2005) is a widely used metric for well-ordered property evaluation. It matches the unigram and calculates a matching score by considering the harmonic mean of precision and recall simultaneously. Note that, recall is weighted more heavily than precision.
- *Consensus-based Image Description Evaluation (CIDEr)* (Vedantam et al., 2015) is widely used in evaluating the quality of image/video captioning. It measures the sentence-level cosine similarity between the candidate caption and human description.

For multi-label classification, the protein-centric maximum F-Score ($F_{max}$) (Gligorijević et al., 2021) is used for evaluation. $F_{max}$ can be calculate as:

$$F_{max} = \max_{\sigma \in [0,1]} \left\{ \frac{2 \times \text{Precision}(\sigma) \times \text{Recall}(\sigma)}{\text{Precision}(\sigma) + \text{Recall}(\sigma)} \right\} \tag{7}$$

where $\sigma$ is a maximum threshold. $F_{max}$ is the maximum F-Score in the range $[0, 1]$.

## B  TRAINING DETAILS OF DOWNSTREAM TASKS

We first introduce four tasks and datasets:

- *Protein Fold Classification* (Hou et al., 2018) is an important task for protein structure understanding with a total of 1195 fold classes. The dataset contains 16,712 proteins, and provides three evaluation scenarios: Fold, Superfamily, and Family. We follow (Hermosilla et al., 2021) to split the dataset into `Train` / `Val` / `Test_Fold` / `Test_Superfamily` / `Test_Family` with 12,312 / 736 / 718 / 1254 / 1272 proteins, respectively.
- *Enzyme Reaction Classification* (Hermosilla et al., 2021) aims to classify the function of the protein based on enzyme reaction. The dataset contains more than 37k proteins with 384 four-level Enzyme Commission (EC) classes (Webb, 1992). The annotated data have 29,215 proteins for training, 2,562 for validation, and 5,651 for testing.

|  (a) BLEU | (b) METEOR | (c) ROUGE-L | (d) CIDEr |

Figure 6: Ablation on the ProteinCap dataset size. We train P2T-GPT on the subsets of ProteinCap under 20%, 40%, 60%, and 80% partitions, respectively. We report BLEU@1-4, METEOR, ROUGE-L, and CIDEr on the entire test set. It shows that 80% training data is basically enough for the task.

- *Gene Ontology Term Prediction* (Gligorijević et al., 2021) can be seen as a multi-label classification task. It contains three subtasks: biological process (BP), molecular function (MF), and cellular component (CC) with 1,943, 489, and 320 classes, respectively. According to Gligorijević et al. (2021), the dataset is split into training, validation, and test sets with 29,898 / 3,322 / 3,415 proteins.

- *Enzyme Commission Number Prediction* (Gligorijević et al., 2021) is also a multi-label classification task which is different from enzyme reaction classification. The dataset has a total of 538 EC categories, and we follow (Gligorijević et al., 2021) that split into 15,550 / 1,729 / 1,919 proteins for `Train`/`Val`/`Test`.

For the single-label classification task (fold classification and function classification), we train the first 150 epochs with an initial learning rate of $1 \times 10^{-4}$, then train another 50 epochs with a learning rate of $1 \times 10^{-5}$.

For gene ontology term prediction, we train 50 epochs with the learning rate of $1 \times 10^{-4}$. For the enzyme commission number prediction task, we train 600 epochs in total. We set the initial learning rate of $1 \times 10^{-4}$ and decay twice at the 200th epoch and 400th epoch. Binary cross-entropy loss is used for both gene ontology term and enzyme commission number prediction.

## C DIAGNOSTIC EXPERIMENTS ON DATASET SIZE

We explore whether the scale of the ProteinCap dataset is sufficient for protein captioning. We further train P2T-GPT under partition protocols of 20%, 40%, 60%, and 80%. Results are shown in Figure 6. We observe that the performance improves slightly as the proportion of the dataset rises from 80% to 100%. This suggests that our ProteinCap dataset is enough for protein captioning under the current setting.

## D ARCHITECTURE OF THE BASELINE MODEL

We illustrate the detailed architecture of the self-attention baseline in Figure 7 and compare it with our method. Hyperparameters are the same as those of P2T-GPT in Section 4.3 of the main paper.

## E LIMITATIONS

To collect sufficient data for protein captioning, our protein encoder extracts protein representation at the sequence level. As mentioned by (Alexander et al., 2009), proteins with similar amino acid sequences may fold into very different 3D geometry structures, leading to different functions or attributes. Therefore, P2T-GPT may be not that effective for those sequences. This can be addressed by integrating 3D modeling into the protein encoder. However, due to the insufficiency of 3D protein data with textural descriptions, we cannot conduct 3D protein captioning by far. It can be studied in the future.

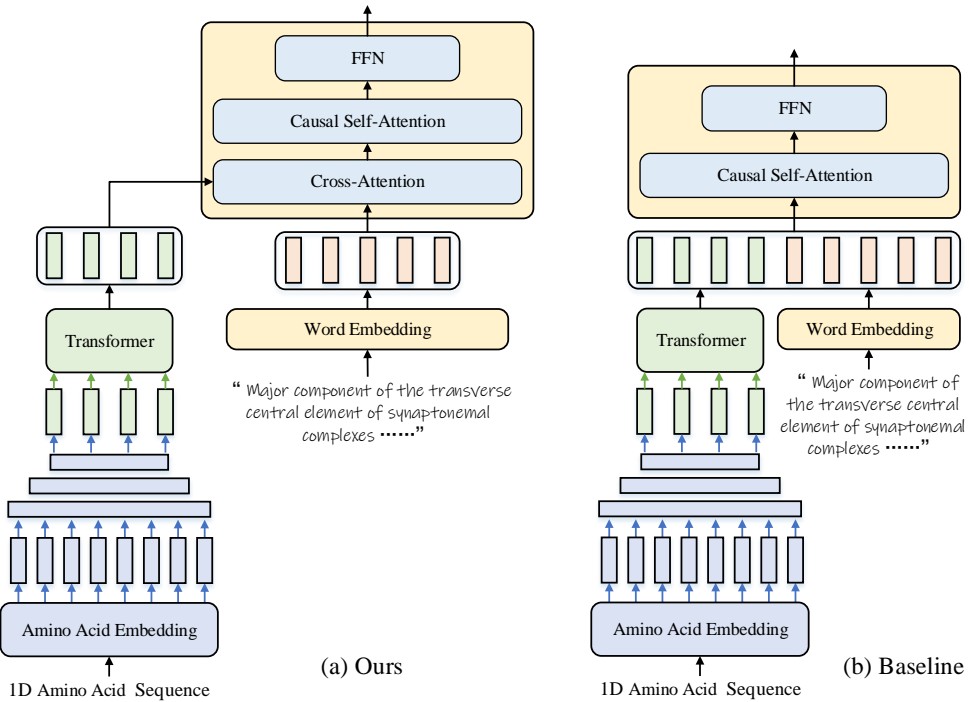

Figure 7: Comparison of our model and self-attention baseline. (a) Ours. (b) Baseline.

## F ADDITIONAL QUALITATIVE VISUALIZATION AND PROTEIN QUESTION ANSWERING.

In Figure 8-10, we provide more qualitative results on ProteinCap-$\alpha$, ProteinCap-$\beta$, and ProteinCap-$\gamma$, respectively. We highlight correct and incorrect descriptions in blue (green) and red. It shows that, compared to the self-attention baseline, P2T-GPT effectively improves the captioning accuracy, especially for long amino acid sequences.

We also provide visual results of cross-attention maps in Figure 11-13. This demonstrates that the high strength of the activation part in the cross-attention map usually corresponds to the important description in the caption.

Furthermore, we provide an additional conversation example in Figure 14.

**Protein ID: Q24JX9**

**Ground Truth：** In the hair cortex , hair keratin intermediate filaments are embedded in an interfilamentous matrix , consisting of hair keratin-associated proteins ( KRTAP ) , which are essential for the formation of a rigid and resistant hair shaft through their extensive disulfide bond cross-linking with abundant cysteine residues of hair keratins . The matrix proteins include the high-sulfur and high-glycine-tyrosine keratins ( By similarity ).

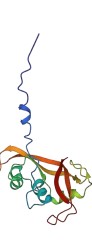

**P2T-GPT :** In the hair cortex , hair keratin intermediate filaments are embedded in an interfilamentous matrix , consisting of hair keratin-associated proteins ( KRTAP ) , which are essential for the formation of a rigid and resistant hair shaft through their extensive disulfide bond cross-linking with abundant cysteine residues of hair keratins . The matrix proteins include the high-sulfur and high-glycine-tyrosine keratins.

**Baseline:** Inhibits peptide that binds to the cell ( By similarity ).

**Protein ID: Q9D244**

**Ground Truth：** Ribonuclease which shows a preference for the pyrimidines uridine and cytosine . Has potent antibacterial activity against a range of Gram-positive and Gram-negative bacteria , including P.aeruginosa , A.baumanii , M.luteus , S.aureus , E.faecalis , E.faecium , S.saprophyticus and E.coli . Causes loss of bacterial membrane integrity , and also promotes agglutination of Gram-negative bacteria ( By similarity ) . Probably contributes to urinary tract sterility . Bactericidal activity is independent of RNase activity ( By similarity ).

**P2T-GPT :** Ribonuclease which shows a preference for the pyrimidines uridine and cytosine . Has potent antibacterial activity against a range of Gram-positive and Gram-negative bacteria , including P.aeruginosa , A.baumanii , M.luteus , S.aureus , E.faecalis , E.faecium , S.saprophyticus and E.coli . Causes loss of bacterial membrane integrity , and also promotes agglutination of Gram-negative bacteria . Probably contributes to urinary tract sterility . Bactericidal activity is independent of RNase activity.

**Baseline:** Involved in the hydrolysis of the cell.

**Protein ID: P26525**

**Ground Truth：** NDH-1 shuttles electrons from an unknown electron donor , via FMN and iron-sulfur ( Fe-S ) centers , to quinones in the respiratory and/or the photosynthetic chain . The immediate electron acceptor for the enzyme in this species is believed to be plastoquinone . Couples the redox reaction to proton translocation , and thus conserves the redox energy in a proton gradient.

**P2T-GPT :** NDH-1 shuttles electrons from NAD ( P ) H : plastoquinone , via FMN and iron-sulfur ( Fe-S ) centers , to quinones in the photosynthetic chain and possibly in a chloroplast respiratory chain . The immediate electron acceptor for the enzyme in this species is believed to be plastoquinone . Couples the redox reaction to proton translocation , and thus conserves the redox energy in a proton gradient.

**Baseline:** NDH-1 shuttles electrons from NADH , via FMN and iron-sulfur ( Fe-S ) centers , to quinones in the respiratory chain . Couples the redox reaction to proton translocation ( for every two electrons transferred , four hydrogen ions are translocated across the cytoplasmic membrane ).

Figure 8: Qualitative comparison. We compare P2T-GPT with the baseline on the ProteinCap-$\alpha$ test set. The blue and red colors indicate correct and incorrect descriptions. The green color indicates the correct descriptions of the baseline method.

**Protein ID: Q7MGC2**

**Ground Truth**：Catalyzes the phosphorolysis of diverse nucleosides , yielding D-ribose 1-phosphate and the respective free bases . Can use uridine , adenosine , guanosine , cytidine , thymidine , inosine and xanthosine as substrates . Also catalyzes the reverse reactions.

**P2T-GPT** ：Catalyzes the phosphorolysis of diverse nucleosides , yielding D-ribose 1-phosphate and the respective free bases . Can use uridine , adenosine , guanosine , cytidine , thymidine , inosine and xanthosine as substrates . Also catalyzes the reverse reactions.

**Baseline:** Catalyzes the hydrolysis of the degradation of a beta-alanine.

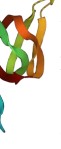

**Protein ID: Q6AJR0**

**Ground Truth**：A translational regulator that binds mRNA to regulate translation initiation and/or mRNA stability . Usually binds in the 5'-UTR at or near the Shine-Dalgarno sequence preventing ribosome-binding , thus repressing translation . Its main target seems to be the major flagellin gene , while its function is anatagonized by FliW.

**P2T-GPT** ： A A translational regulator that binds mRNA to regulate translation initiation and/or mRNA stability . Usually binds in the 5'-UTR at or near the Shine-Dalgarno sequence preventing ribosome-binding , thus repressing translation . Its main target seems to be the major flagellin gene , while its function is anatagonized by.

**Baseline:** The be a growing fatty to the presence of fatty acid biosynthesis . The binds to the synthesis of the P , it binds to the T site of the formation of the formation of a specificity of the midpoint of the long to the cell.

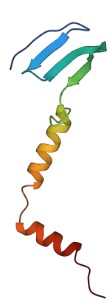

**Protein ID: P63121**

**Ground Truth**：Retroviral proteases have roles in the processing of the primary translation products and the maturation of the viral particle . Endogenous Pro proteins may have kept , lost or modified their original function during evolution.

**P2T-GPT** ： Retroviral proteases have roles in processing of the primary translation products and the maturation of the viral particle . Endogenous Pro proteins may have kept , lost or modified their original function during evolution . This endogenous protein has retained most of the characteristics of characteristics.

**Baseline:** Catalyzes the formation of the RNA.

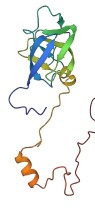

**Protein ID: P63121**

**Ground Truth**：The pigment-dispersing hormone causes the migration of the distal retinal pigment into the proximal end of the pigment chromatophore cells and thus decreases the amount of light entering the retinulas . May also function as a neurotransmitter and/or neuromodulator.

**P2T-GPT** ： The pigment-dispersing hormone causes the migration of the distal retinal pigment into the proximal end of the pigment chromatophore cells and thus decreases the amount of light entering the retinulas . May also function as a neurotransmitter and/or neuromodulator ( By similarity ).

**Baseline:** May be a role in the cell and in the cell , and proteins , and the twin-arginine of the cell , and peptide of the could the cell.

Figure 9: Qualitative comparison. We compare P2T-GPT with the baseline on the ProteinCap-$\beta$ test set. The blue and red colors indicate correct and incorrect descriptions. The green color indicates the correct descriptions of the baseline method.

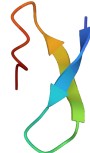

**Protein ID: P30266**

**Ground Truth:** Possesses antifungal activity sensitive to inorganic cations.

**P2T-GPT:** Possesses antifungal activity sensitive to inorganic cations.

**Baseline:** Toxin that inhibits both growth . This toxin inhibits potassium channels.

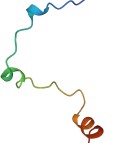

**Protein ID: O33672**

**Ground Truth:** Acts as a pheromone , induces cells to develop competence for genetic transformation.

**P2T-GPT:** Acts as a pheromone , induces cells to develop competence for genetic transformation.

**Baseline:** Acts as an pheromone , induces cells to develop competence for genetic transformation , and malonyl thioester to 4 Cys , and recombination

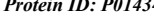

**Protein ID: P01434**

**Ground Truth:** Binds to muscle nicotinic acetylcholine receptor ( nAChR ) and inhibit acetylcholine from binding to the receptor , thereby impairing neuromuscular transmission.

**P2T-GPT:** Binds to muscle nicotinic acetylcholine receptor ( nAChR ) and inhibit acetylcholine from binding to the receptor , thereby impairing neuromuscular transmission.

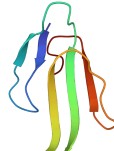

**Baseline:** Produces peripheral paralysis by blocking neuromuscular transmission at the postsynaptic site . Binds to the nicotinic acetylcholine receptor ( nAChR ) . This toxin blocks peripheral paralysis by cobrotoxin-b ( By similarity ).

**Protein ID: P69173**

**Ground Truth:** C protein is one of the proteins involved in the production and packaging of viral single-stranded DNA

**P2T-GPT:** C protein is one of the proteins involved in the production and packaging of viral single-stranded DNA

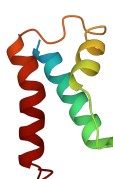

**Baseline:** Regulates viral DNA release of the proteins involved in the production and packaging of viral single-stranded DNA

**Protein ID: C0HKR1**

**Ground Truth:** This hormone , released from cells in the corpora cardiaca , causes release of diglycerides from the fat body and stimulation of muscles to use these diglycerides as an energy source during energy-demanding processes.

**P2T-GPT:** This hormone , released from cells in the corpora cardiaca , causes release of diglycerides from the fat body and stimulation of muscles to use these diglycerides as an energy source during energy-demanding processes.

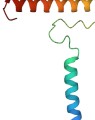

**Baseline:** Antimicrobial peptide that seems to be involved in secretion from root species ( 50 ) biosynthesis of the competition at them.

Figure 10: Qualitative comparison. We compare P2T-GPT with the baseline on the ProteinCap-$\gamma$ test set. The blue and red colors indicate correct and incorrect descriptions. The green color indicates the correct descriptions of the baseline method.

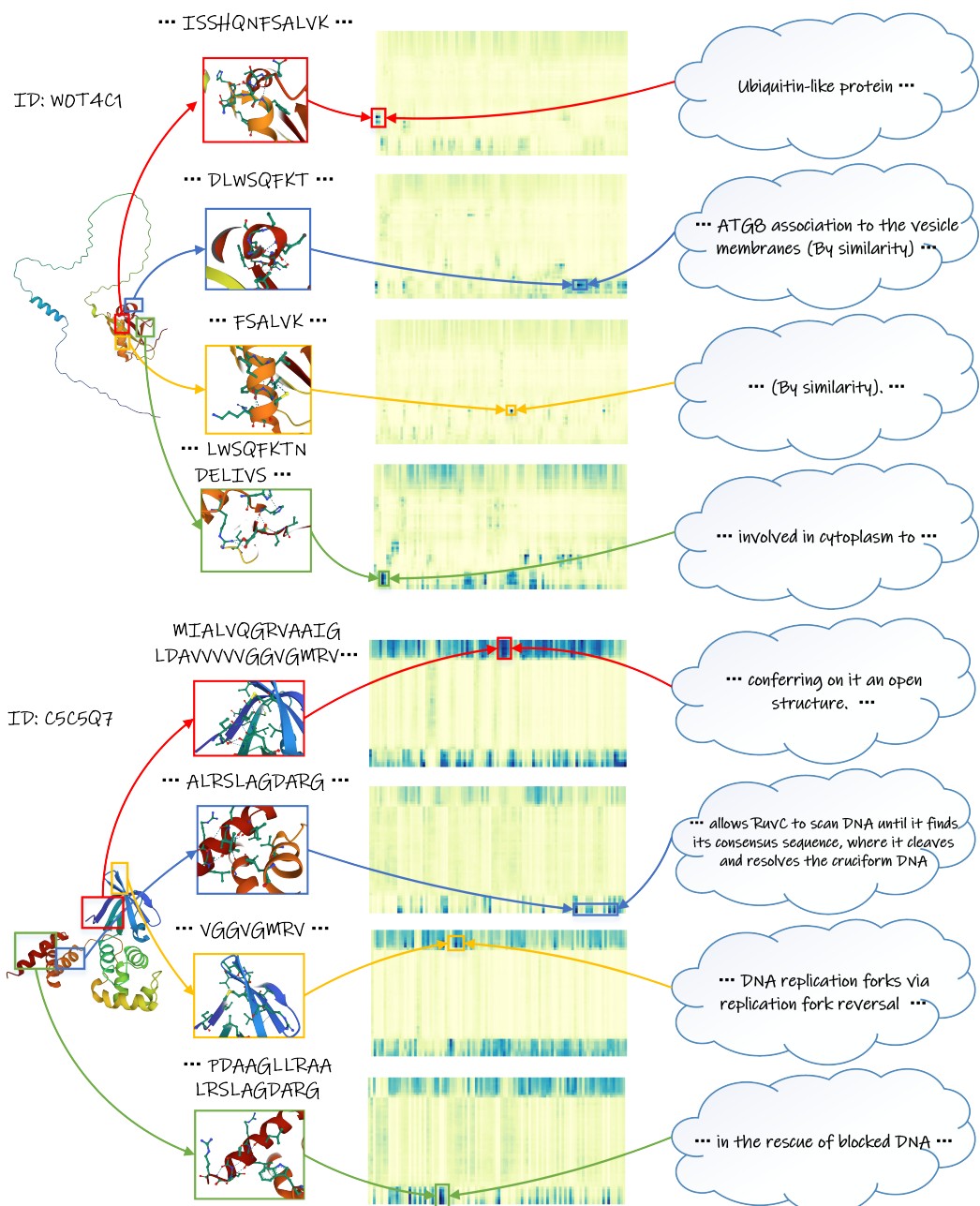

Figure 11: Visualization of the cross-attention maps on the ProteinCap-α test set. We provide the amino acid sequence and description corresponding to the high-strength activation section in the cross-attention map. The results are from two different proteins (WOT4C1 and C5C5Q7).

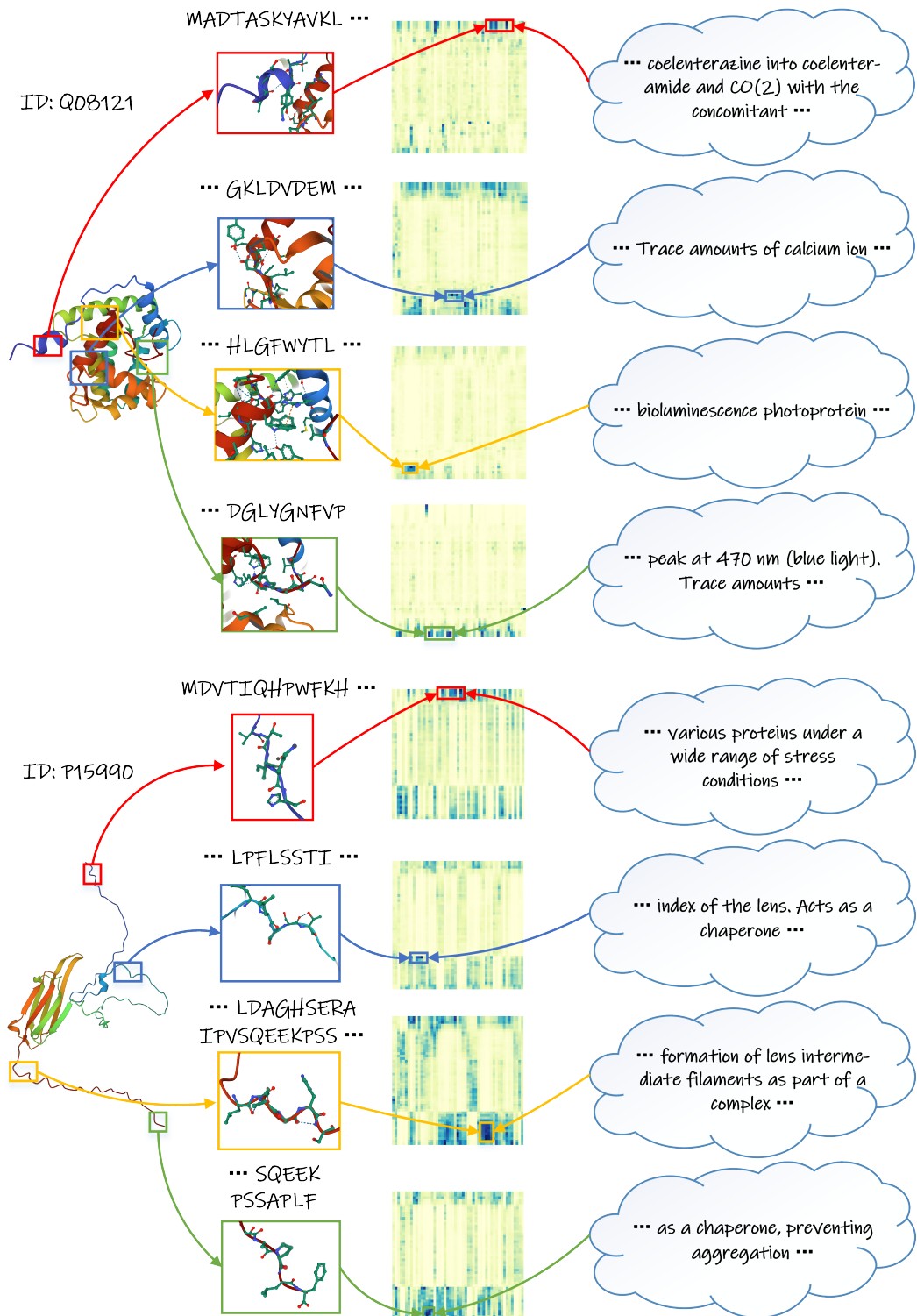

Figure 12: Visualization of the cross-attention maps on the ProteinCap-$\beta$ test set. We provide the amino acid sequence and description corresponding to the high-strength activation section in the cross-attention map. The results are from two different proteins (Q08121 and P15990).

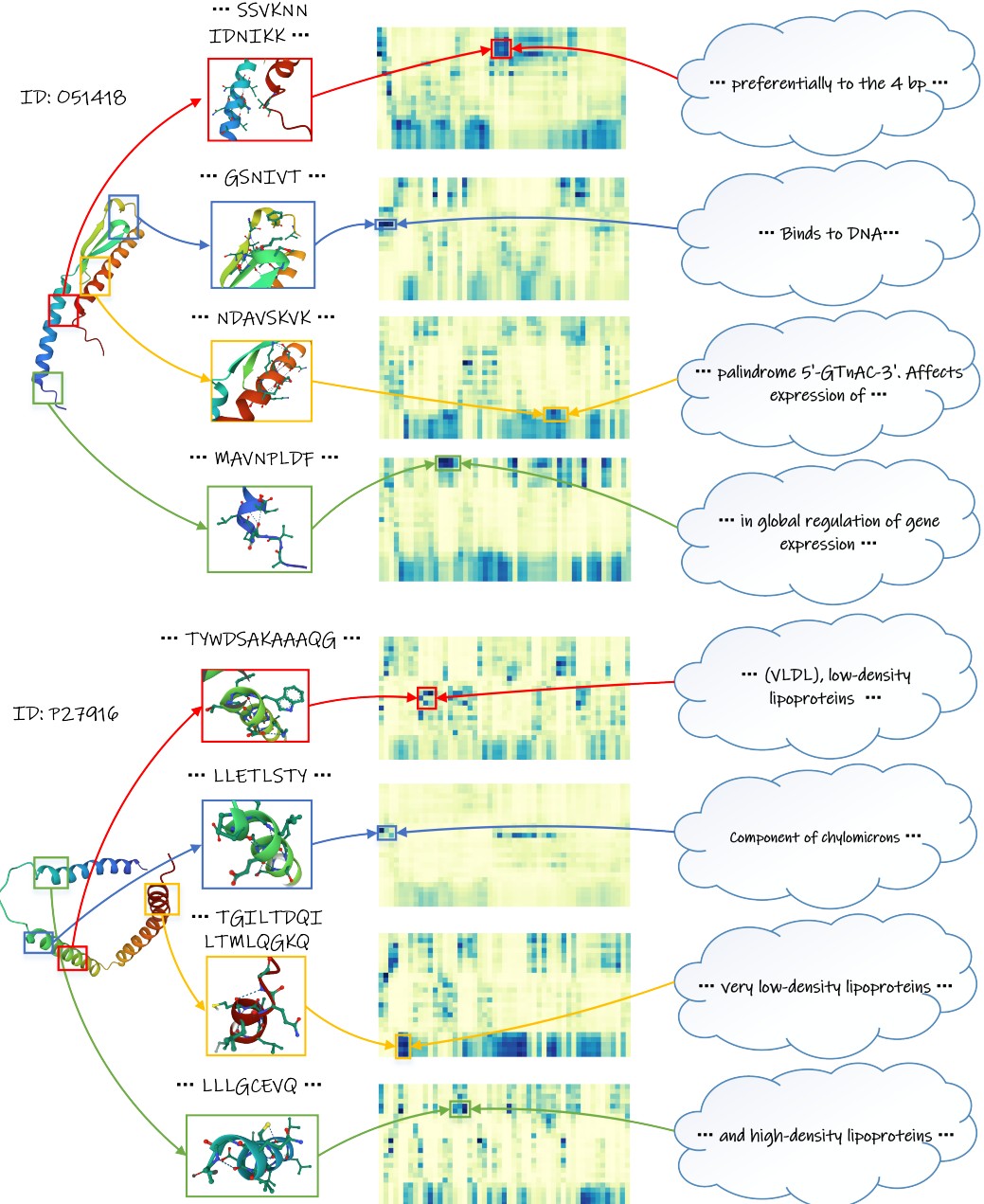

Figure 13: Visualization of the cross-attention maps on the ProteinCap-$\gamma$ test set. We provide the amino acid sequence and description corresponding to the high-strength activation section in the cross-attention map. The results are from two different proteins (O51418 and P27916).

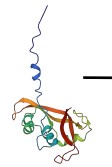

*Protein ID Q9D244*

**P2T-GPT (Ours)**

*Here is the captioning of a protein sequence: "Ribonuclease which shows a preference for the pyrimidines uridine and cytosine. Has potent antibacterial activity against a range of Gram-positive and Gram-negative bacteria , including P.aeruginosa , A.baumanii , M.luteus , S.aureus , E.faecalis , E.faecium , S.saprophyticus and E.coli . Causes loss of bacterial membrane integrity , and also promotes agglutination of Gram-negative bacteria . Probably contributes to urinary tract sterility . Bactericidal activity is independent of RNase activity."*

*Please tell me are there any therapeutic or clinical applications for harnessing the antibacterial properties of this ribonuclease, particularly in the context of urinary tract infections or bacterial infections more broadly?*

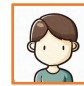

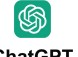

**ChatGPT**

*The ribonuclease's potent antibacterial properties could have therapeutic potential for treating urinary tract infections and other bacterial infections. Its ability to target a broad spectrum of bacteria, including drug-resistant strains, suggests it may be valuable in developing new antimicrobial treatments, although further research and clinical studies are needed to explore its clinical applications fully.*

*How does this ribonuclease potentially contribute to urinary tract sterility, and what are the factors influencing its bactericidal activity independent of RNase activity?*

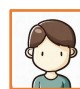

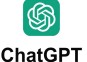

**ChatGPT**

*This ribonuclease likely contributes to urinary tract sterility by exhibiting potent antibacterial activity, causing bacterial membrane damage, and promoting agglutination of Gram-negative bacteria. Its bactericidal activity independent of RNase activity suggests that factors other than RNA degradation are responsible for its antimicrobial effects, which require further investigation.*

*How does this ribonuclease exhibit a preference for pyrimidines uridine and cytosine, and what is the significance of this specificity in its function?*

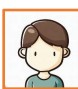

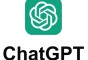

**ChatGPT**

*The mechanism underlying the ribonuclease's preference for pyrimidines uridine and cytosine remains unclear from the provided information. However, this specificity is significant as it may allow the ribonuclease to selectively target RNA molecules containing these pyrimidines, potentially affecting essential cellular processes and contributing to its antibacterial activity.*

Figure 14: A conversation that ChatGPT (OpenAI, 2022) answers the question based on the generated protein captioning.

