# OpenReview forum: "Protein Captioning: Bridging the Gap between Protein Sequences and Natural Languages"
_ICLR.cc/2024/Conference — ICLR 2024 Conference Withdrawn Submission_

### Official Review · Reviewer_gDSy · 2023-10-30

**Soundness:** 3 good
**Presentation:** 3 good
**Contribution:** 3 good
**Rating:** 6
**Confidence:** 4

**Summary:**

This paper introduces a model for “protein captioning,” analogous to image captioning in vision, where a protein sequence serves as input to a model that outputs a natural language description of the protein that captures attributes. The model trains a sequence encoder consisting of convolutional+downsampling layers to get an encoding of the amino acid sequence, followed by a transformer to capture longer-range sequence relationships. The caption decoder is trained to output a natural-language description of the protein. A cross-attention mechanism is employed to attend sequence regions with components of the description. The model is evaluated on the captions themselves (UniProt function descriptions) and as a pretraining step on the downstream tasks of fold classification, enzyme reaction classification, GO term prediction, and EC number prediction.

**Strengths:**

The paper addresses an interesting and challenging problem of predicting protein attributes from sequence. The model proposed makes use of modeling approaches from the protein representation learning and NLP fields to learn useful descriptions of proteins from sequence alone. Descriptions generated achieve high fidelity with ground truth and performance is considered for descriptions up to 100 tokens over a large vocabulary.

The cross-attention for sequence and description is an interesting component that links sequence regions to specific components to the protein function, and this choice is also well supported in experiments. The qualitative and quantitative results show improvements over a self-attention mechanism over sequence and description together.

**Weaknesses:**

ProtNLM [Gane et al] should be addressed as this model also outputs descriptions from protein amino acid sequences. The use of this model in the UniProt curation pipeline calls this tasks "sequence annotation,” but is the same as what is here called protein captions. The model is developed to produce annotations/captions that include protein names, EC number, protein family, and organism classification.

[Gane et al]: Gane et al. ProtNLM: Model-based Natural Language Protein Annotation (2022), (available at https://www.uniprot.org/help/ProtNLM).

Model only allows for proteins length up to 200 amino acids. This represents only about 30% of sequences in the most recent (2023_04) release of SwissProt.

Random split for train, validate, test is used, not dependent on for example, sequence similarity, organism classification, homology, or description/functional similarity. Often the challenge of protein annotation or description is applying methods to proteins that are quite different than those seen in training. Otherwise, alignment-based homology can be used to transfer knowledge. Such a “traditional” baseline or a more careful splitting of the dataset would strengthen the results of this model.

The question answering task is under-developed in the main text, with results limited mostly to the appendix. The contribution is difficult to evaluate given its exposition in the submission. While it is definitely an interesting avenue for future work, the utility of the description as input to conversational agents cannot be evaluated here robustly. Are the responses from the agent useful, correct, etc and how should these be evaluated? Do the natural text descriptions produce more useful or correct responses compared to other prompts? These would be interesting avenues for evaluating this component, but at the moment it serves more as a motivation for future work rather than a contribution in its own right for this paper.

**Questions:**

While I think this paper is well written and generally addresses well and carefully this challenging and important task, I have the following questions and minor points of lack of clarity. I hope the authors can help address these.

Modeling:
Q1] Why was the CNN+Transformer approach chosen to encode sequence? Were other methods considered for the sequence encoding step prior to the transformer, e.g. pretrained protein langues models such as ESM, ProtT5, or others? For downstream tasks (Table 4), the use of ESM encodings are shown to vastly improve performance. Would this also be case for the captioning task itself?

Q2] How were model hyperparameters chosen?

Dataset:
Q3] What is meant by dataset contains proteins with “comprehensive properties”?

Q4] The dataset partitions are split by description length in addition to sequence length. Which model should be applied to a protein needing captioning that has a sequence between 20 and 200 amino acids, ProteinCap-alpha or ProteinCap-beta? How do models trained on one partition and evaluated on another compare? What about an un-partitioned model where all sequences are used?

Q5] Can you specify the number of proteins removed at each filtering step? The sequence length would cut the SwissProt dataset down by ~70% (N=~175,00), but what about the description length? It’s not clear if the remaining 94,454 proteins are a result of description-length filtering or missing function description. If the former, would e.g. truncating the description allow for expanding the dataset for a more robust model?

Minor comments/Writing
The first paragraph says that “existing protein representation learning methods usually focus on one or more a few specific and individual classification tasks.” Can you rephrase this to explain what you mean? Representation learning techniques such as protein language models are trained in an unsupervised way to reconstruct sequences. Do you mean that these works focus their evaluation on a few tasks? E.g. TAPE [Rao et al 2019] propose 5 benchmark tasks for comparing sequence representation methods but does not use these to train models.
[Rao et al 2019]: Rao et al, Evaluating Protein Transfer Learning with TAPE, NeurIPS 2019

Section 4 first paragraph “reasonably” > “reasonable”

---

### Official Review · Reviewer_2Fof · 2023-10-30

**Soundness:** 2 fair
**Presentation:** 2 fair
**Contribution:** 2 fair
**Rating:** 3
**Confidence:** 3

**Summary:**

Unlike classical protein processing tasks that mainly involves naive classification tasks, the authors propose to challenge machine to understand protein in a text generation format, which is commonly in the natural language processing. The authors introduce the task as 'protein caption' and the model need to directly generate verbalized plain text output to describe the input protein, including its functions, atrributes, etc.

The 'caption' dataset is collected from an existed protein dataset followed by some dataset filtering and vocabulary building processes, which make the dataset more compatible with langauge generation techniques.

To solve the 'caption' tasks, the authors propose a model called P2T-GPT. It contains a protein encoder which consists of several convolution layers and a language decoder where classical causal language masking is applied to train.

On the 'caption' dataset, the authors report some generation evaluation metrics, e.g. BLEU and BERTScore, to show how much their model can achive and compare different attention mechanism inside their model designs.
And when trained firstly on the 'caption' dataset, the encoder can better work on other classical downstream tasks, i.e. completing the classical classification tasks using the globle token representations of the encoder output.

**Strengths:**

- The proposed task setting is valuable. It challenges machine to understand protein by directly generating descriptions. As the descriptions can contain more information than classification labels and generating descriptions is more user-friendly, aiming at this setting is appreciatable.

- It is excited to see the protein caption can be treated as a pretraining objective. The results show the pretrained encoder can better work on classical classification tasks to boost existed SOTA to higher performances. These results are insightful.

**Weaknesses:**

Although the setting is promising, the authors do not reach a satisfactory completition.
- As the authors say caption can describe the protein from diverse perspectives, i.e. functions, attributes. The authors should convincely demonstrate how the dataset covers such desirable properties. What perspectives those caption contain. More detailed data statistics is nesessary. In the currect version, I do not think the collected dataset real reveal the advantages of protein caption.
- The data collection process is too simple and trival to reflect the chanllenges and values of protein caption. The authors transfer an existed dataset as their caption dataset. Although some data filtering processes are applied, those techniques are common in NLP community. It is necessary to collect the dataset with strong biology knowledge ground. Then it can really challenge the AI for understanding protein.

 The authors only report generation metrics, i.e. BLEU and to show how much score their generation mdoels can achieve, and some cases studies about how it models can prompt ChatGPT. The results are not informative.
- I suggest the authors to back-transfer the plain text output to classification labels and calculate metrics such as accuracy or recall and compare with baseline methods that trained in a clasification manner. In-context learning and other techniques in NLP can make this. By such, I can better access how well the generation task format can help to complete those complex protein processing tasks.
- The authors also argue their models can incorporate with ChatGPT in a conversation way. But again, the results are not informative. The case analysis lacks illustrations.

The proposed model is trival. Protein encoder, some attention mechanism and causal decoder are widely-exploited in currect community. Therefore, I do not think this paper contributes to protein understanding with novel models.

**Questions:**

Please address my concerns list in the Weakness Part.

---

### Official Review · Reviewer_gqUf · 2023-10-31

**Soundness:** 2 fair
**Presentation:** 2 fair
**Contribution:** 2 fair
**Rating:** 3
**Confidence:** 5

**Summary:**

The paper you described introduces Protein Captioning and presents the Protein-to-Text Generative Pre-trained Transformer (P2T-GPT) as a method to translate protein sequences into natural language descriptions.

**Strengths:**

1.The paper presents the task of Protein Captioning, which offers a flexible way for protein analysis.

2.The paper proposes the P2T-GPT model, which is designed to convert amino acid sequences into natural language text.

3.The effectiveness of P2T-GPT is demonstrated through experiments on the ProteinCap dataset.

**Weaknesses:**

1.The training data mentioned in the paper consists of only a few thousand (or a few tens of thousand) data points. This amount of data seems relatively small. Moreover, the constructed metrics 'primaryAccession' (protein ID), 'length' (protein length), 'sequence' (amino acid sequence), and 'function' (textual description of protein function, attributes, or other information) are overly naive.

2.It is not clear from the text whether the decoder part is fine-tuned using a pre-trained GPT model, as there doesn't appear to be any explicit statement about this in the paper.

3.In the paper, convolutional neural networks are used in the structural diagrams to capture local structural information in the amino acid sequences. However, the role of this component is not clearly demonstrated in the experimental section. For cross-modal tasks, an intuitive approach could be to use pre-trained models like ESM-2 in the encoder part (or other cross-modal models such as Q-former and Flamingo). Given the limited amount of training data in the paper, it might have been more beneficial to use pre-trained models as demonstrated in Table 4. The authors should consider providing an explanation for not adopting this approach.

4.When conducting baseline comparisons, the paper introduces one method for comparison. However, there is another common comparison method, which involves jointly constructing a corpus of amino acid sequences and text descriptions for training (e.g., fine-tuning on Galactica).

5.In Section 3.2 of the paper, there should be a brief introduction to ProteinCap-α, ProteinCap-β, and ProteinCap-γ.

**Questions:**

see above

---

### Official Review · Reviewer_CGFR · 2023-11-01

**Soundness:** 3 good
**Presentation:** 3 good
**Contribution:** 2 fair
**Rating:** 6
**Confidence:** 4

**Summary:**

In this paper, the authors propose the P2T-GPT framework for protein captioning that uses natural language to describe proteins. The P2T-GPT is an encoder-decoder language model consisting of a protein encoder and a causal captioning decoder. To train P2T-GPT, the authors collected the ProteinCap dataset with 94,454 protein-text pairs. Experimental results show that P2T-GPT can provide accurate text descriptions for protein analysis.

**Strengths:**

The paper is written clearly and is well-structured; the motivation is intuitive and the method is easy to follow.

The author collected large-scale protein-text pairs data, which enhances the reliability and generalizability of the proposed P2T-GPT.

Ablation studies on dataset size and model attention mechanism demonstrate the effectiveness of the model design and the proposed dataset.

Extensive experiments on diverse downstream protein recognition tasks are relatively comprehensive. The additional attention analysis also increases the model's interpretability.

**Weaknesses:**

No positional embeddings are mentioned; hence the spatial structure information of the protein might be lost.

The results in downstream tasks show only marginal improvement compared to ESM-2, which may be related to the loss of structure information. A more detailed explanation and deeper investigation should be provided for further research on model design.

**Questions:**

I do not have additional questions at this stage.